# Neutralizing Antibody Responses Elicited by Inactivated Whole Virus and Genetic Vaccines against Dominant SARS-CoV-2 Variants during the Four Epidemic Peaks of COVID-19 in Colombia

**DOI:** 10.3390/vaccines10122144

**Published:** 2022-12-14

**Authors:** Diego A. Álvarez-Díaz, Ana Luisa Muñoz, María T. Herrera-Sepúlveda, Pilar Tavera-Rodríguez, Katherine Laiton-Donato, Carlos Franco-Muñoz, Héctor Alejandro Ruiz-Moreno, Marisol Galindo, Jenssy D. Catama, Andrea Bermudez-Forero, Marcela Mercado-Reyes

**Affiliations:** 1Genómica de Microorganismos Emergentes, Dirección de Investigación en Salud Pública, Instituto Nacional de Salud, Bogotá 111321, Colombia; 2Facultad de Ciencias Biología, Universidad Nacional de Colombia, Bogotá 111321, Colombia; 3Fundación Banco Nacional de Sangre Hemolife, Bogotá 110911, Colombia; 4Dirección de Investigación en Salud Pública, Instituto Nacional de Salud, Bogotá 111321, Colombia; 5Grupo de Parasitología, Dirección de Investigación en Salud Pública, Instituto Nacional de Salud, Bogotá 111321, Colombia

**Keywords:** SARS-CoV-2, Gamma, Delta, Mu, Omicron, neutralizing antibodies, BNT162b2, CoronaVac, ChAdOx1, Ad26.COV2.S, vaccines

## Abstract

Several SARS-CoV-2 variants of concern (VOC) and interest (VOI) co-circulate in Colombia, and determining the neutralizing antibody (nAb) responses is useful to improve the efficacy of COVID-19 vaccination programs. Thus, nAb responses against SARS-CoV-2 isolates from the lineages B.1.111, P.1 (Gamma), B.1.621 (Mu), AY.25.1 (Delta), and BA.1 (Omicron), were evaluated in serum samples from immunologically naïve individuals between 9 and 13 weeks after receiving complete regimens of CoronaVac, BNT162b2, ChAdOx1, or Ad26.COV2.S, using microneutralization assays. An overall reduction of the nAb responses against Mu, Delta, and Omicron, relative to B.1.111 and Gamma was observed in sera from vaccinated individuals with BNT162b2, ChAdOx1, and Ad26.COV2.S. The seropositivity rate elicited by all the vaccines against B.1.111 and Gamma was 100%, while for Mu, Delta, and Omicron ranged between 32 to 87%, 65 to 96%, and 41 to 96%, respectively, depending on the vaccine tested. The significant reductions in the nAb responses against the last three dominant SARS-CoV-2 lineages in Colombia indicate that booster doses should be administered following complete vaccination schemes to increase the nAb titers against emerging SARS-CoV-2 lineages.

## 1. Introduction

Five vaccines against the severe acute respiratory syndrome coronavirus 2 (SARS-CoV-2) are authorized in Colombia; by the second week of July 2022, around 86 million vaccine doses had been administered. The inactivated whole-virus vaccine CoronaVac (Sinovac Life Sciences, Beijing, China), and the BNT162b2 (Pfizer-BioNTech), an mRNA-based vaccine that encodes the SARS-CoV-2 full-length spike (S) gene, are the two main vaccines administered in the country, each one accounting for 30% of the doses. In addition, the ChAdOx1 (Oxford-AstraZeneca) and Ad26.COV2.S-Janssen (Johnson & Johnson) vaccines which are based on replication-incompetent adenoviral vectors, expressing a variant of the SARS-CoV-2 S protein, account for 15.5% and 8.5% of the administered doses, respectively. Finally, the mRNA-1273 vaccine (Moderna), another mRNA-based vaccine that encodes the SARS-CoV-2 full-length S gene of SARS-CoV-2, accounts for 14.2% of the doses administered in the country [1]. However, around 36 million people have received the complete scheme of vaccination, which is two doses for CoronaVac, BNT162b2, ChAdOx1, or Moderna, and one dose for Ad26.COV2.S, while only 9.3 million have received a booster dose with a homologous or heterologous vaccine [2,3].

The routine genomic surveillance of SARS-CoV-2 in Colombia allowed the identification of variants of interest (VOI) and concern (VOC) that co-circulate in the country with evidence of escape to neutralizing antibodies (nAb) generated by the BNT162b2 vaccine or natural infection [4,5]. The first two COVID-19 epidemic peaks in Colombia were dominated by B.1 non-VOC/VOI lineages (B.1.111, B.1.420, and B.1) and occurred from June to September 2020 and December 2020 to January 2021, respectively, while between April to August 2021 the third epidemic peak was dominated by Gamma (P.1) and Mu (B.1.621) lineages, representing 25% and 49% of the total cases [4]. These two variants were gradually displaced by Delta lineages from July to December 2021. Soon afterward, the arrival of Omicron before the end of December 2021, during the fourth epidemic peak, resulted in the total displacement of Delta lineages by the end of January 2022 [6].

The Gamma variant was the first VOC detected in the country. It has 12 mutations in the S protein, including critical substitutions on its receptor binding domain (RBD) (K417N, E484K, and N501Y) associated with an increment in the binding affinity to the human ACE2-receptor [6] and reduced levels of neutralizing antibodies [4,7,8,9]. The Mu variant contains a distinctive profile of mutations in the S protein (D614G, D950N, E484K, ins145N, N501Y, P681H, R346K, T95I, Y144T, Y145S) [10], involved in the resistance to vaccine-elicited or natural infection-elicited antibodies [4,11]. Furthermore, almost one hundred Delta variant (B.1.617) sublineages were detected in Colombia, five of them (AY.20, AY.25, AY.26, AY.3, AY.5) with higher frequency. The dominant sublineage, AY.25, also carries multiple mutations at the S protein (D614G, del157/158, D950N, L452R, P681R, T19R, T478K, V1065L) associated with immune evasion [12,13,14]. Finally, 52 Omicron sublineages have been detected in the country; however, BA.1 was mainly associated with the fourth epidemic peak in Colombia. The BA.1 sublineage is characterized by a profile of 32 mutations in the S protein, including at least three deletions (DEL69/70, DEL143/145, and DEL212), and four mutations of interest (K417N, S477N, N501Y, and P681H). Curiously, the E484K mutation which has been considered an important determinant of escape from vaccine-elicited nAbs, is absent in the Delta and Omicron variants [15].

According to a systematic review of phase III vaccine efficacy studies, the efficacy against symptomatic COVID-19 after complete schemes of BNT162b2, mRNA-1273, ChAdOx1, Ad26.COV2.S or CoronaVac was approximately 95.4%, 93.6%, 74.1%, 58.63%, and 69.4%, respectively. Furthermore, the same study found that the efficacy of these vaccines against severe COVID-19 was around 85.35%, 99.1%, 100%, 85.4%, and 100% [3]. However, the emergence of SARS-CoV-2 VOCs and VOIs carrying genetic markers at the S protein associated with greater transmissibility rate and resistance to nAbs at the end of December 2020 challenged this scenario. Several studies around the world evidenced a substantial fold decrease in the geometric mean nAb titer (GMT) against different VOCs/VOIs, relative to ancestral SARS-CoV-2 lineages in individuals with complete regimens of the vaccines currently approved or authorized in Colombia [3]. Furthermore, a study on nAb responses in BNT162b2-vaccinated in Colombia evidenced a robust reduction in the neutralization of Mu, a VOI first detected in the Caribbean region of the country [10], by 75.7- and 17.7-fold, relative to B.1.111 and Gamma, respectively [4], which was comparable to reports from Asia and Europe [11,16]. Hence, at the end of December 2021 the World Health Organization (WHO) issued an interim statement on booster doses for COVID-19 vaccination in light of the evidence of waning protection from the primary vaccination series [17].

As these reductions in nAbs responses ranged from minimal up to total escape from vaccine nAbs depending on the SARS-CoV-2 variant and the population evaluated [3,18]; it is important to characterize the profile of vaccine nAbs in the context of the circulating SARS-CoV-2 variants in each country. Hence, we used a microneutralization assay with infectious viruses to determine the nAb responses against B.1.111, Gamma, Mu, Delta, and Omicron in individuals fully vaccinated with four out of the five vaccines authorized in Colombia, altogether accounting for 86% of the doses administered: BNT162b2, CoronaVac, ChAdOx1 and Ad26.COV2.S.

## 2. Materials and Methods

### 2.1. Patients and Samples

This was a non-probabilistic, consecutive cross-sectional study of four Colombian cohorts included in the National Vaccination Plan during the prioritization phase [19]. The first three cohorts received a vaccination schedule of two doses at four-week intervals of BNT162b2 (Pfizer), ChAdOx1 (AstraZeneca), or CoronaVac (Sinovac). The fourth cohort consisted of those vaccinated with one dose of Ad26.COV2.S (Janssen). Due to the limited availability of individuals vaccinated with mRNA-1273 (Moderna) at the sampling time, this cohort was not included in the study. Women and men between 18 and 81 years old who were immunologically naïve to COVID-19 were included. Individuals with a previous or current SARS-CoV-2 infection during clinical follow-up, or those with the presence of total antibodies against SARS-CoV-2 at the time of the first dose of vaccine administration, were excluded. Serum samples were collected from immunologically naïve individuals at 13 weeks after receiving the first dose of BNT162b2 (*n* = 31; three males and 28 females, age range, 23–62 years), CoronaVac (*n* = 30; nine males and 21 females; age range 18–58 years) and ChAdOx1 (*n* = 31; 14 males and 17 females; age range 50–81 years). While samples from patients vaccinated with Ad26.COV2.S (*n* = 26; 10 males and 16 females; age range 21–61 years) were collected 20 weeks after receiving one dose of this vaccine. A blood sample (7 mL) was collected from each participant by venipuncture. The blood samples were centrifuged. Then 2 mL of serum was stored in vials and frozen at −70 °C until processing. Serum samples were collected between June 2021 and May 2022. For the sample sub-set of BNT162b2 vaccinated individuals, we used the same patients’ sera and data used previously to compare preliminary results of nAb responses against B.1.111, Mu, and Gamma with new data on different vaccines and SARS-CoV-2 variants [4].

### 2.2. Virus Isolation

Nasopharyngeal swab samples collected during the routine genomic surveillance of SARS-CoV-2 with positive qRT-PCR for SARS-CoV-2, complete genome sequencing, and PANGO lineage assignment B.1.111 (EPI_ISL_526971), P.1/Gamma (EPI_ISL_2500971), B.1.621/Mu (EPI_ISL_1821065), AY.25.1/Delta (EPI_ISL_7314401) and BA.1/Omicron (EPI_ISL_9433093), were selected for virus isolation and used for inoculation of Vero E6 monolayers as described previously [4]. All assays with infectious SARS-CoV-2 viruses were performed in a biocontainment laboratory level 3.

### 2.3. Microneutralization and Binding Antibody Assays

Microneutralization assays with infectious viruses were performed in Vero E6 monolayers as described [5], by incubating 120 mean tissue culture infectious doses (TCID50) of SARS-CoV-2 working stocks (passage 2) isolated and propagated on Vero E6 cells belonging to the lineages B.1.111, P.1 (Gamma), B.1.621 (Mu), AY.25.1 (Delta), and BA.1 (Omicron) with two-fold serial dilutions (1:10 to 1:2560) of sera from vaccine-naïve volunteers. The cytopathic effect was examined on the fifth dayof incubation, and the mean neutralization titer (MN50) was calculated by the Reed–Muench method [20]. The absence of total (IgG/IgM) anti-S antibodies before vaccination was verified using the SARS-CoV-2 Total assay (COV2T, Siemens Healthcare Diagnostics Inc., New York, NY, USA) (Appendix A). The absence of IgG anti-nucleoprotein antibodies during clinical follow-up of individuals vaccinated with BNT162b2, ChAdOx1, Ad26.COV2.S was verified using the qualitative ELISA ID Screen SARS-CoV-2-N IgG Indirect (ID Vet, Montpellier, France), following the manufacturer’s instructions (Appendix A). Subsequently, the concentration of anti-spike IgG antibodies was tested using the SARS-CoV-2 IgG assay (sCOVG, Siemens Healthcare Diagnostics Inc., New York, NY, USA) on the ADVIA Centaur XPT platform (Siemens) (Appendix A). The cut-off index value was defined as reactive ≥ 1.0 U/mL. Index values were expressed in binding antibody units per milliliter (BAU)/mL using the conversion factor (WHO standard) of 21.8, as determined for the Siemens assays, which was the BAU cut-off value [21].

### 2.4. Statistical Analysis

As data did not follow a Gaussian distribution (D’Agostino-Pearson test, α cut-off at 0.05), the statistical differences between the Anti-S IgG antibody titers and mean neutralization titer (MN50) for BNT162b2, CoronaVac, ChAdOx1, and Ad26.COV2.S against each SARS-CoV-2 variant were determined using the Kruskal–Wallis test, followed by Dunn’s post hoc test for multiple comparisons, where a *p*-value of <0.05 was statistically significant. For samples with BAU titers above the linearity of the method (>2180 BAU/mL), an arbitrary value of 3270 BAU/mL was assigned. Also, an arbitrary value of 5 was assigned to samples with MN50 < 10. Non-parametric Spearman’s rank correlation was performed to calculate the correlation between geometric means of binding and neutralization antibodies for each vaccine. Statistical analyses were performed using GraphPad PRISM 9.3.1 (GraphPad Software, San Diego, CA, USA).

## 3. Results

### 3.1. Vaccine Elicited Anti-S IgG Titers Correlate with Neutralizing Antibody Titers

No significant differences were observed between the binding anti-S IgG titers elicited by Ad26.COV2.S, ChAdOx1, and CoronaVac. However, the titers elicited by BNT162b2 were significantly higher than the other three vaccines (Figure 1). Remarkably, 14 out of the 31 individuals vaccinated with Pfizer had antibody titers above the linearity of the method (>2180 BAU/mL). Hence, because an arbitrary value of 3270 BAU/mL was assigned to those samples, it is probable that their binding anti-S IgG titers were underestimated. Neutralizing antibody titers elicited by BNT162b2, CoronaVac, ChAdOx1, and Ad26.COV2.S against B.1.111, Mu, and P.1, were correlated with total binding anti-S IgG antibody titers. Nevertheless, nAbs elicited by BNT162b2 and Ad26.COV2.S against the Omicron variant showed no correlation with binding Anti-S IgG antibodies. The same absence of correlation was observed with the nAbs elicited by BNT162b2 against the Delta variant (Table 1).

### 3.2. Reduced nAb Responses against Mu, Delta, and Omicron Variants in Vaccinated Individuals with BNT162b2, CoronaVac, ChAdOx1, and Ad26.COV2.S

Microneutralization assays with sera from individuals vaccinated with BNT162b2, ChAdOx1, and Ad26.COV2.S revealed an overall reduction of the GMT against B.1.621 (Mu), AY.25.1 (Delta), and BA.1 (Omicron), relative to B.1.111 and P.1 (Gamma) (Figure 2a). In contrast, the GMT for CoronaVac was significantly lower only for Mu and Omicron (Figure 2b).

Thus, while the overall GMTs elicited by the four vaccines against B.1.111 and Gamma ranged between 51.7 to 401.3, and 65 to 139 respectively, for Mu, Delta, and Omicron ranged between 7.2 to 27, 15.8 to 47, and 15.6 to 62, respectively (Table 2). These reductions were remarkable against Mu and Omicron in individuals vaccinated with BNT162b2 and CoronaVac, followed by the nAb responses against Delta in individuals vaccinated with ChAd0x1-s and Ad26.COV2.S, as the GMTs and seropositivity rates were the lowest among all the vaccines and variants tested (Table 2).

Neutralizing antibody titers from all the tested vaccines were uniformly distributed against AY.25.1 (Delta) because, when compared, only slight significant differences were observed in the GMTs between CoronaVac and Ad26.COV2.S (Figure 2a). The highest nAb response was induced by BNT162b2 against B.1.111 (Figure 2b), followed by ChAdOx1 and Ad26.COV2.S against Gamma (Figure 2b). Finally, the ChAdOx1 vaccine showed the best performance against Mu, as the GMTs and seropositivity rate against this variant were higher in contrast to the other vaccines (Figure 2a,b).

## 4. Discussion

Massive vaccinations against SARS-CoV-2 around the world began in December 2020, reducing the rate of COVID-19-related hospitalizations and deaths [21,22]. However, the emergence of VOC and VOI challenged the effectiveness of the most widely used COVID-19 vaccines, and in consequence, surveillance of their performance became a priority. This surveillance should include studies across different populations to estimate the global effectiveness of SARS-CoV-2 vaccines [22] as well as to reconsider vaccination programs and redesign existing vaccines.

An indicator of the protective effect of current vaccines on emerging variants is to evaluate their neutralizing capacity. We previously reported the neutralizing responses of BNT162b2-vaccinated individuals against Mu and Gamma [4]. However, a more complete picture of vaccine-elicited nAb responses against the main SARS-CoV-2 variants in Colombia is required. Thus, at least to our knowledge, this is the first assessment of the nAb responses against Gamma, Mu, Delta, and Omicron (SARS-CoV-2 lineages with widespread circulation in Colombia) in individuals with complete schemes of BNT162b2, CoronaVac, ChAdOx1, and Ad26.COV2.S, the four main COVID-19 vaccines administered in the country. These results show that inactivated whole virus and mRNA-based vaccines induce different levels of neutralizing antibodies against SARS-CoV-2 variants with high epidemiologic impacts, such as the VOI Mu, Delta, and Omicron variants. In this study, the nAb responses after complete schemes of BNT162b2, CoronaVac, ChAdOx1, or Ad26.COV2.S were significantly lower against the VOI Mu and the VOCs Delta and Omicron, relative to B.1.111 and Gamma, which could imply an overall reduced efficacy of these COVID-19 vaccines.

Besides, we observed differences between the nAb titers induced by different vaccine platforms. In particular, the Gamma, Mu, and Omicron variants showed less resistance to nAbs elicited by adenoviral vector-based COVID-19 vaccines, ChAdOx1, or Ad26.COV2.S, while nAb responses against Delta were higher in CoronaVac-vaccinated patients.

The stabilization state of the Spike protein can explain differences in the neutralizing antibody titers elicited by the inactivated whole virus and genetic vaccines (mRNA and Adenovirus-vector vaccines). For example, the manufacturing process of the CoronaVac vaccine includes beta-propiolactone to inactivate viral particles and several purification steps which increase S1 shedding from the S trimer on SARS-CoV-2 virions which affect the conformation of S and its presentation to the immune system. Such factors may contribute to variations in the efficacies reported for this vaccine [3,23].

On the other hand, the mRNA vaccine BNT162b2 introduced stabilizing mutations intended to prevent conformational instability of the spike protein and subsequent unwanted shedding of S1 [23]. However, our results suggest that this sequence optimization does not prevent the escape of several SARS-CoV-2 variants from nAbs elicited by this vaccine.

Furthermore, the adenoviral vaccine Ad26.COV2.S contains S-stabilizing mutations to keep the prefusion structure of S, and ChAdOx1 lacks S-stabilizing mutations. Nonetheless, both platforms use signal peptide sequences at the N-terminus of S; Ad26.COV2.S use the original SARS-CoV-2 S protein signal sequence, while ChAdOx1 uses an extended form of the tissue plasminogen activator (tPA) signal sequence upstream of the original S protein signal sequence [23]. Hence, while the higher nAb responses elicited by genetic vaccines probably rely on the presentation of a stable form of the spike protein as an antigen to the immune system, the best performance of adenovirus-vector vaccines relative to mRNA vaccines could be due to differences in the stability of antigen expression.

In line with our findings, studies in individuals vaccinated with BNT162b2, ChAdOx1, and CoronaVac evidenced reduced nAb responses against Gamma, Delta, and Omicron, compared to an ancestral variant carrying the D614G mutation, although these reductions were remarkable for Omicron [3,8,12,14,24]. Additionally, individuals with two doses of the BNT162b2 vaccine showed a decrease in the neutralizing antibody titers against Gamma, Alpha, Delta, and Mu variants after one month of vaccination, with a noticeable reduction in the titers against Delta and Mu variants [25]. Similarly, studies on patients with two doses of BNT162b2 or CoronaVac reported no or low nAb responses against Omicron relative to the ancestral variant [18,26]. Moreover, several studies evidenced that nAb titers decreased dramatically after six months for Gamma, Alpha, Delta, Mu, and Omicron [25,26].

Genetic, epidemiologic, and host factors can contribute to the notorious ability of the variants to escape from nAbs. Although more than 30 mutations have been described in the spike gene of Omicron, a profile of mutations is shared with Mu (T95I, R346K, N501Y, and P681H) and Delta (L452R, T478K, and P681H) [27].Multiple studies support the role of some of these mutations on immune evasion and greater transmission fitness. The T95I mutation combined with G142D was associated with higher viral loads and predicted to reduce the neutralization capacity of post-vaccinated sera by impairing the nAb binding to the epitopes 4–8 (7L2E) and 4A48 (7C2L) in the N-terminal domain (NTD) [28]. The R346K mutation was also associated with resistance to class 2 nAbs by reducing the binding ability to the RBD [29,30].

Finally, the L452R, T478K, and N501Y mutations were associated with nAbs resistance and higher binding affinity to the ACE2 receptor, consequently increasing viral transmissibility [28,31,32,33]. Hence, it is possible that the reduced nAb responses against Mu, Delta, and Omicron found in our work can be partially explained by these mutations.

On the other hand, despite multiple studies showing differential vaccine-elicited nAb responses against SARS-CoV-2 variants, the impact of nAb titer on the clinical outcome is not fully understood. Andrews et al. 2022 reported significant waning vaccine efficacy over time against Delta and Omicron (the most predominant variants lately) after two doses of BNT162b2 or ChAdOx1. This study found a reduction of BNT162b2 effectiveness against infection with Delta from 90.9% at four weeks from the second dose, to 62.7% at 25 weeks. However, the effectiveness against infection with Omicron decreased from 65.5% at four weeks from the second dose to just 8.8% at 25 weeks. Moreover, the effectiveness of ChAdOx1 against infection with Delta decreased from 82.8% at four weeks to 43.5% at 25 weeks, while the effectiveness against Omicron decreased from 48.9% at four weeks to no effect at 20 weeks [34], which is consistent with our neutralization results and with earlier reports.

Although high nAb levels correlate positively with protection against COVID-19 and vaccine effectiveness [22,35], the protective titer of neutralizing antibodies remains to be determined. Lau et al. reported that a hemagglutination inhibition antibody titer of 1:40 protects from infection with influenza [36]. Moreover, Gharbharan et al., suggest that a titer of 1:80 or higher after the convalescent plasma transfusion for the treatment of COVID-19 inhibits viral growth in vitro by 95% [37]. For these reasons, we hypothesize that a titer of 1:80 or higher may protect against progression to severe disease, which, for public health reasons, should be achieved preferably using vaccination. However, some vaccine recipients with poor or nonexistent nAb responses may develop protective cellular immunity after SARS-CoV-2 exposure. Furthermore, mutations associated with nAb resistance do not always result in a prominent evasion of a cell-specific response, which may diminish the severity of the infection. Hence, it is still necessary to do research that investigates both cellular and humoral immunity to provide a more comprehensive profile of the immune response to novel variants [38,39].

Altogether, the evidence points to considerable drops in both nAb responses and vaccine efficacy against SARS-CoV-2 emerging variants with a high impact on public health. However, this study has several limitations, including a small sample size with demographic heterogeneity, the absence of samples from the pediatric or adolescent population, and a lack of information on the contribution of B or T cell responses against the variants studied.

## 5. Conclusions

In conclusion, this study shows that the three last SARS-CoV-2 variants with the most recent circulation in the country have a greater ability to escape from neutralizing antibodies induced by the BNT162b2, ChAdOx1, and CoronaVac vaccines. Hence, the low nAbs responses against Mu, Delta, and Omicron support the need for booster doses to improve the protection against SARS-CoV-2 infection or severe COVID-19. However, it remains to be studied in what proportion this will improve the magnitude of nAb titers and clinical efficacy. Thus, other non-pharmacological measures should be retained in the vaccinated population.

Finally, the continuous emergence of SARS-CoV-2 variants with a high ability to elude vaccine-elicited nAbs emphasizes the importance of continuing surveillance programs that include studies across different populations, monitoring nAb responses after booster doses, and natural infection with emerging SARS-CoV-2 variants to estimate the global effectiveness of current SARS-CoV-2 vaccines. It is also critical to develop new vaccine strategies to enhance cross-protective immunity against new potential variants, as well as to target viral factors involved in cell infection to improve vaccination strategies.

## Figures and Tables

**Figure 1 vaccines-10-02144-f001:**
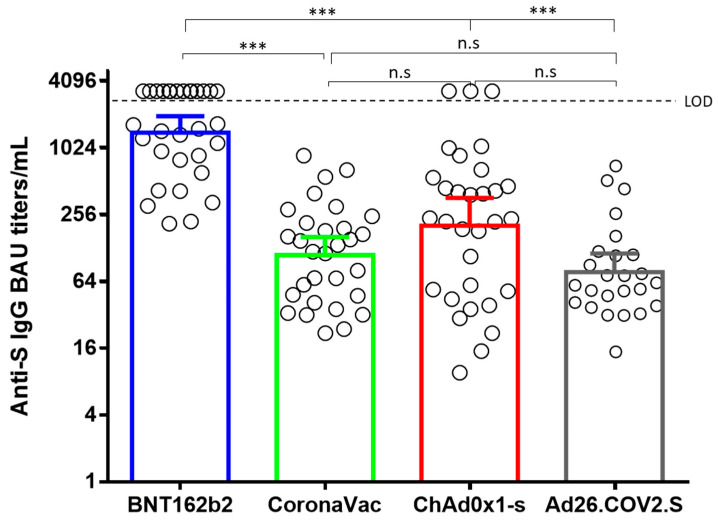
Comparison of geometric means of binding Anti-S IgG antibody titers. BAU: binding antibody units. ***: *p* < 0.0001. n.s: no significant.

**Figure 2 vaccines-10-02144-f002:**
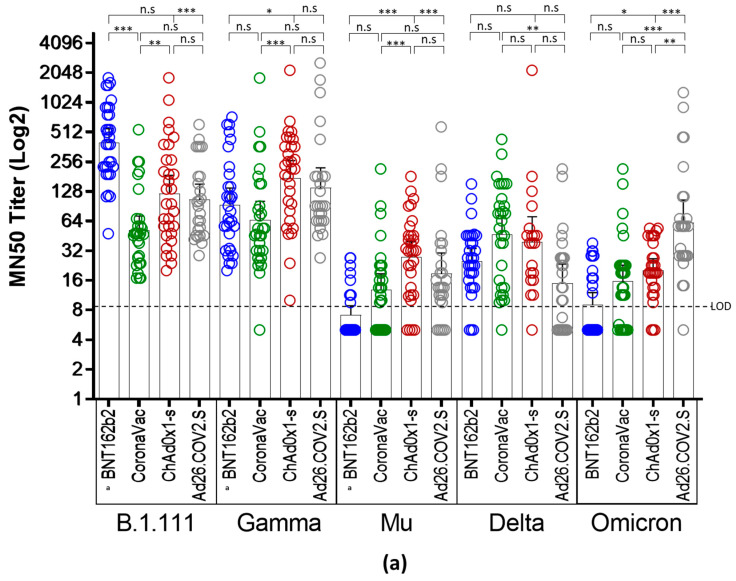
Comparison of neutralizing activity in BNT162b2, CoronaVac, ChAdOx1, and Ad26.COV2.S -vaccinated individuals by SARS-CoV-2 variants. (**a**) Comparison of GMTs between vaccines against each SARS-CoV-2 variant. (**b**) Comparison of GMTs between SARS-CoV-2 variants for each vaccine. *: *p* < 0.01, **: *p* < 0.001, ***: *p* < 0.0001. n.s: no significant. LOD: limit of detection. ^a^: data of nAb titers from BNT162b2-vaccinated individuals were reported previously [4] and re-analyzed considering a limit of detection of 1:10.

**Table 1 vaccines-10-02144-t001:** Comparison of neutralizing and binding Anti-S IgG antibody titers.

Vaccine	GMBA Titers (BAU/mL)/(95% CI)	Comparison		Spearman r *	*p*-Value
BNT16b2	1353 (959.9–1907)	^a^ MN50 B.1.111	^a^ Anti-S IgG titer BAU/mL	0.674	<0.0001
		^a^ MN50 Gamma (P.1)	^a^ Anti-S IgG titer BAU/mL	0.533	0.002
		^a^ MN50 Mu (B.1.621)	^a^ Anti-S IgG titer BAU/mL	0.511	<0.003
		MN50 Delta (AY.25.1)	Anti-S IgG titer BAU/mL	0.258	0.161
		MN50 Omicron (BA.1)	Anti-S IgG titer BAU/mL	−0.089	0.6302
CoronaVac	111.3 (76.31–162.4)	MN50 B.1.111	Anti-S IgG titer BAU/mL	0.709	<0.0001
		MN50 Gamma (P.1)	Anti-S IgG titer BAU/mL	0.453	0.0118
		MN50 Mu (B.1.621)	Anti-S IgG titer BAU/mL	0.627	<0.0002
		MN50 Delta (AY.25.1)	Anti-S IgG titer BAU/mL	0.580	0.0008
		MN50 Omicron (BA.1)	Anti-S IgG titer BAU/mL	0.478	0.0075
ChAd0x1-s	198 (113.8–345)	MN50 B.1.111	Anti-S IgG titer BAU/mL	0.656	<0.0001
		MN50 Gamma (P.1)	Anti-S IgG titer BAU/mL	0.773	<0.0001
		MN50 Mu (B.1.621)	Anti-S IgG titer BAU/mL	0.723	<0.0001
		MN50 Delta (AY.25.1)	Anti-S IgG titer BAU/mL	0.8	<0.0001
		MN50 Omicron (BA.1)	Anti-S IgG titer BAU/mL	0.575	0.0013
Ad26.COV2.S	78.36 (53.2–115.4)	MN50 B.1.111	Anti-S IgG titer BAU/mL	0.625	0.0008
		MN50 Gamma (P.1)	Anti-S IgG titer BAU/mL	0.725	<0.0001
		MN50 Mu (B.1.621)	Anti-S IgG titer BAU/mL	0.597	0.0016
		MN50 Delta (AY.25.1)	Anti-S IgG titer BAU/mL	0.779	<0.0001
		MN50 Omicron (BA.1)	Anti-S IgG titer BAU/mL	0.389	0.054

GMBA: geometric mean of binding antibody titers. * Two-tailed. ^a^: binding and neutralizing antibody titers from BNT162b2 vaccinated individuals were reported previously [4], and re-analyzed considering a limit of detection of 1:10.

**Table 2 vaccines-10-02144-t002:** Summary of vaccine antibody responses against SARS-CoV-2 variants.

	Vaccine	B.1.111 (Ancestral Virus)	P.1 (Gamma)	B.1.621 (Mu)	B.1.617.2 (Delta AY.25.1)	BA.1 (Omicron)
Seropositive rate * % (no. positive/total)	BNT162b2	^a^ 100% (31/31)	^a^ 100% (31/31)	^a^ 32.2% (10/31)	90.3% (28/31)	41.2% (13/31)
	CoronaVac	100% (30/30)	100% (30/30)	63.3% (19/30)	96.6% (29/30)	73.3% (22/30)
	ChAd0x1-s	100% (31/31)	100% (31/31)	87% (27/31)	74.2% (23/31)	89,3% (25/28)
	Ad26.COV2.S	100% (26/26)	100% (26/26)	80.7% (21/26)	65.4% (17/26)	96.2% (25/26)
Geometric mean TCID50 titer (95% CI)	BNT162b2	^a^ 401.3 (288.0–559.2)	^a^ 94.02 (63.61–139)	^a^ 7.2 (5.8–8.9)	25.3 (19–34)	25.4 (18.9–33.9)
	CoronaVac	51.7 (42.6–124.5)	65.7 (42.2–102.1)	12.9 (8.9–18.7)	47.1 (30.4–72.9)	15.6 (10.7–22.9)
	ChAd0x1-s	121.6 (79.2–186.8)	176.7 (118–264.6)	27.6 (19.2–39.5)	39.2 (21.6–71.4)	20.3 (15.4–26.7)
	Ad26.COV2.S	106.1 73.9–152.2)	139.8 (87.4–223.6)	18.8 (11.6–30.5)	15.8 (9.6–23.6)	62.1 (36.7–105.1)
GMT Fold decrease relative to B.1.111	BNT162b2	-	4.26	55.71	15.8	15.8
	CoronaVac	-	0.78	4.0	1.1	3.3
	ChAd0x1-s	-	0.91	4.4	3.1	5.9
	Ad26.COV2.S	-	0.75	5.6	6.7	1.7

* MN titer ≥ 10. ^a^: data from nAb titers from BNT162b2-vaccinated individuals were reported previously [4] and re-analyzed considering a limit of detection of 1:10.

## Data Availability

All supporting data are included in this manuscript.

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
