# Peer review of "Neutralizing Antibody Responses Elicited by Inactivated Whole Virus and Genetic Vaccines against Dominant SARS-CoV-2 Variants during the Four Epidemic Peaks of COVID-19 in Colombia"

_vaccines, 2022, doi:10.3390/vaccines10122144_

Round 1

Reviewer 1 Report

Alvarez Dia et al evaluated the neutralizing antibody responses elicited by inactivated whole virus and mRNA vaccines against dominant SARS-CoV-2 variants during the four epidemic peaks of COVID-19 in Colombia.

The authors explored several SARS-CoV-2 variants of concern (VOC) and interest (VOI) that co-circulate in Colombia and determined the neutralizing antibody (nAb) responses against SARS-CoV-2 isolates from the line-24 ages B.1.111, P.1 (Gamma), B.1.621 (Mu), AY.25.1 (Delta), and BA.1 (Omicron) using 25 serum samples from immunologically naïve individuals receiving 26 complete regimens CoronaVac, BNT162b2, ChAdOx1, or Ad26.COV2.

They described an overall reduction of the nAb responses against Mu, Delta, and Omicron, relative to 28 B.1.111 and Gamma in sera from vaccinated individuals with BNT162b2, ChAdOx1, 29 and Ad26.COV2.S.

Data obtained by the team are interesting and explore a large number of variants, including Mu.

But, the discussion part deserves to be improved.

First, line 213, the authors cite a reduced nAb response against Gamma, Mu, Delta and Omicron, whereas no reduce response against Gamma was observed (cf. results).

The impact of each important mutation of the different variants could be discussed to explain the differences observed between the variants and between the vaccines. All of the information discussed don’t fit well together.

Minor comments.

The Figure 2a is difficult to read.

Author Response

Response to Reviewer 1 Comments

Reviewer comments:

  1. The discussion part deserves to be improved.

Response 1: We agree with the suggestion, all the discussion was edited. (with changes marked Pages 7-10, Line 221-326).

  1. Discussion

Massive vaccinations against SARS-CoV-2 around the world began in December 2020, reducing the rate of COVID-19-related hospitalizations and deaths [21,22]. However, the emergence of VOC and VOI challenged the effectiveness of the most widely used COVID-19 vaccines, and in consequence, surveillance of their performance became a priority. This surveillance should include studies across different populations to estimate the global effectiveness of SARS-CoV-2 vaccines [1], as well the reconsideration of vaccination programs and the redesign of existing vaccines.

An indicator of the protective effect of current vaccines on emerging variants is to evaluate their neutralizing capacity. Thus, at least to our knowledge, this is the first assessment of the nAb responses against Gamma, Mu, Delta, and Omicron (SARS-CoV-2 lineages with widespread circulation in Colombia) in individuals with two doses of BNT162b2, CoronaVac, ChAdOx1, or Ad26.COV2.S, the four main COVID-19 vaccines  administered in the country. These results show that inactivated whole virus and mRNA-based vaccines induce different levels of neutralizing antibodies against SARS-CoV-2 variants with high epidemiologic impacts such as the VOI Mu, Delta, and Omicron. In this study, the nAb responses after complete schemes of BNT162b2, CoronaVac, ChAdOx1, or Ad26.COV2.S were significantly lower against the VOI Mu and the VOCs Delta and Omicron, relative to B.1.111 and Gamma, which could imply an overall reduced efficacy of these COVID-19 vaccines.

Besides, we observed differences between the nAb titers induced by different vaccine platforms. In particular, the Gamma, Mu, and Omicron variants showed less resistance to nAbs elicited by adenoviral vector-based COVID-19 vaccines, ChAdOx1, or Ad26.COV2.S, while nAb responses against Delta were higher in CoronaVac vaccinated patients.

The stabilization state of the Spike protein can explain differences in the neutralizing antibody titers elicited by the inactivated whole virus and genetic vaccines (mRNA and Adenovirus-vector vaccines). For example, the manufacturing process of the CoronaVac vaccine includes beta-propiolactone to inactivate viral particles and several purification steps which increase S1 shedding from the S trimer on SARS-CoV-2 virions which affect the conformation of S and its presentation to the immune system. Factors that may contribute to variations in the efficacies reported for this vaccine [2,3].

On the other hand, the mRNA vaccine BNT162b2 introduced stabilizing mutations intended to prevent conformational instability of the spike protein and subsequent unwanted shedding of S1 [3]. However, our results suggest that this sequence optimization does not prevent the escape of several SARS-CoV-2 variants from nAbs elicited by this vaccine.

Furthermore, the adenoviral vaccine Ad26.COV2.S contains S-stabilizing mutations to keep the prefusion structure of S, and ChAdOx1 lacks S-stabilizing mutations. Nonetheless, both platforms use signal peptide sequences at the N-terminus of S; Ad26.COV2.S use the original SARS-CoV-2 S protein signal sequence, while ChAdOx1, an extended form of the tissue plasminogen activator (tPA) signal sequence upstream of the original S protein signal sequence [3]. Hence, while the higher nAb responses elicited by genetic vaccines probably rely on the presentation of a stable form of the spike protein as an antigen to the immune system, the best performance of adenovirus-vector vaccines relative to mRNA vaccines could be due to differences in the stability of antigen expression. 

In line with our findings, studies in individuals vaccinated with BNT162b2, ChAdOx1, and CoronaVac evidenced reduced nAb responses against Gamma, Delta, and Omicron, compared to an ancestral variant carrying the D614G mutation, although these reductions were remarkable for Omicron [4]. Additionally, individuals with two doses of the BNT162b2 vaccine showed a decrease in the neutralizing antibody titers against Gamma, Alpha, Delta, and Mu variants, after one month of vaccination with a noticeable reduction in the titers against Delta and Mu variants [5].  Similarly, studies on patients with two doses of BNT162b2 or CoronaVac reported no or low nAb responses against Omicron relative to the ancestral variant [6,7]. Additionally, several studies evidenced that nAb titers decreased dramatically after six months for Gamma, Alpha, Delta, Mu, and Omicron [5,7].

Genetic, epidemiologic, and host factors can contribute to the notorious ability of the variants to escape from nAbs. Although more than 30 mutations have been described in the spike gene of Omicron, a profile of mutations is shared with Mu (T95I, R346K, N501Y, and P681H) and Delta (L452R, T478K, and P681H) [8].

Multiple studies support the role of some of these mutations on immune evasion and greater transmission fitness. The T95I mutation combined with G142D was associated with higher viral loads and predicted to reduce the neutralization capacity of post-vaccinated sera by impairing the nAb binding to the epitopes 4-8 (7L2E) and 4A48 (7C2L) in the N-terminal domain (NTD) [9]. The R346K mutation was also associated with resistance to class 2 nAbs by reducing the binding ability to the RBD [10,11].

Finally, the L452R, T478K, and N501Y mutations were associated with nAbs resistance and higher binding affinity to the ACE2, consequently increasing viral transmissibility receptor [9,12-14]. Hence, it is possible that the reduced nAb responses against Mu, Delta, and Omicron found in our work can be partially explained by these mutations.

On the other hand, despite multiple studies showing differential vaccine-elicited nAb responses against SARS-CoV-2 variants, the impact of nAb titer on the clinical outcome is not fully understood. Andrews., et al 2022 reported significant waning vaccine efficacy over time against Delta and Omicron (the most predominant variants lately) after two doses of BNT162b2 or ChAdOx1. This study found a reduction of BNT162b2 effectiveness against infection with Delta from 90.9% at four weeks from the second dose, to 62.7% at 25 weeks. However, the effectiveness against infection with Omicron decreased from 65.5% at four weeks from the second dose to just 8.8% at 25 weeks. Moreover, the effectiveness of ChAdOx1 against infection with Delta decreased from 82.8% at four weeks to 43.5% at 25 weeks, while the effectiveness against Omicron decreased from 48.9% at four weeks to no effect at 20 weeks [15], which is consistent with our neutralization results and with earlier reports.

Although high nAb levels correlate positively with protection against COVID-19 and vaccine effectiveness [1,16], it remains to be determined the protective titer of neutralizing antibodies. Lau et al. reported that a hemagglutination inhibition titer of 1:40 protects from infection with influenza [17].  Moreover, Gharbharan A. et al., suggest that a titer of 1:80 or higher after the convalescent plasma transfusion for the treatment of COVID-19 inhibits viral growth in vitro by 95% [18]. For these reasons, we hypothesize that a titer of 1:80 or higher may protect against progression to severe disease, which, for public health reasons, should be achieved preferably using vaccination.  However, some vaccine recipients with poor or nonexistent nAb responses may develop protective cellular immunity after SARS-CoV-2 exposure. Furthermore, mutations associated with nAb resistance do not always result in a prominent evasion of a cell-specific response, which may diminish the severity of the infection. Hence, it is still necessary to do research that investigates both cellular and humoral immunity to provide a more comprehensive profile of the immune response to novel variants [19,20].

Altogether, the evidence points to considerable drops in both nAb responses and vaccine efficacy against SARS-CoV-2 emerging variants with a high impact on public health. However, this study has several limitations, including a small sample size with demographic heterogeneity, the absence of samples from the pediatric or adolescent population, and a lack of information on the contribution of B or T cell responses against the variants studied

Point 2: First, line 213, the authors cite a reduced nAb response against Gamma, Mu, Delta and Omicron, whereas no reduce response against Gamma was observed (cf. results).

Response 2: We agree with the suggestion, this paragraph of the discussion was edited. (Page 8, Lines 237-240).

In this study, the nAb responses after complete schemes of BNT162b2, CoronaVac, ChAdOx1, or Ad26.COV2.S were significantly lower against the VOI Mu and the VOCs Delta and Omicron, relative to B.1.111 and Gamma, which could imply an overall reduced efficacy of these COVID-19 vaccines.

Point 3: The impact of each important mutation of the different variants could be discussed to explain the differences observed between the variants and between the vaccines. All of the information discussed don’t fit well together.

Response 3: We agree with the suggestion, the discussion was improved. (Page 9, Lines 284-294).

Multiple studies support the role of some of these mutations on immune evasion and greater transmission fitness. The T95I mutation combined with G142D was associated with higher viral loads and predicted to reduce the neutralization capacity of post-vaccinated sera by impairing the nAb binding to the epitopes 4-8 (7L2E) and 4A48 (7C2L) in the N-terminal domain (NTD) [9]. The R346K mutation was also associated with resistance to class 2 nAbs by reducing the binding ability to the RBD [10,11].

Finally, the L452R, T478K, and N501Y mutations were associated with nAbs resistance and higher binding affinity to the ACE2, consequently increasing viral transmissibility receptor [9,12-14]. Hence, it is possible that the reduced nAb responses against Mu, Delta, and Omicron found in our work can be partially explained by these mutations.

Minor comments.

Point 4: The Figure 2a is difficult to read.

Response 4: Figure 2a was intended to highlight the differences in the nAb responses elicited by each vaccine against individual SARS-CoV-2 variants. The full resolution figures are available in this submission.

References (the number of the references in this document differs from the manuscript because the reference manager)

  1. Fukutani, K.F.; Barreto, M.L.; Andrade, B.B.; Queiroz, A.T.L. Correlation Between SARS-Cov-2 Vaccination, COVID-19 Incidence and Mortality: Tracking the Effect of Vaccination on Population Protection in Real Time. Front Genet 2021, 12, 679485, doi:10.3389/fgene.2021.679485.
  2. Fiolet, T.; Kherabi, Y.; MacDonald, C.J.; Ghosn, J.; Peiffer-Smadja, N. Comparing COVID-19 vaccines for their characteristics, efficacy and effectiveness against SARS-CoV-2 and variants of concern: a narrative review. Clin Microbiol Infect 2022, 28, 202-221, doi:10.1016/j.cmi.2021.10.005.
  3. Heinz, F.X.; Stiasny, K. Distinguishing features of current COVID-19 vaccines: knowns and unknowns of antigen presentation and modes of action. NPJ Vaccines 2021, 6, 104, doi:10.1038/s41541-021-00369-6.
  4. Bian, L.; Liu, J.; Gao, F.; Gao, Q.; He, Q.; Mao, Q.; Wu, X.; Xu, M.; Liang, Z. Research progress on vaccine efficacy against SARS-CoV-2 variants of concern. Hum Vaccin Immunother 2022, 18, 2057161, doi:10.1080/21645515.2022.2057161.
  5. Lopera, T.J.; Chvatal-Medina, M.; Florez-Alvarez, L.; Zapata-Cardona, M.I.; Taborda, N.A.; Rugeles, M.T.; Hernandez, J.C. Humoral Response to BNT162b2 Vaccine Against SARS-CoV-2 Variants Decays After Six Months. Front Immunol 2022, 13, 879036, doi:10.3389/fimmu.2022.879036.
  6. Lu, L.; Mok, B.W.; Chen, L.L.; Chan, J.M.; Tsang, O.T.; Lam, B.H.; Chuang, V.W.; Chu, A.W.; Chan, W.M.; Ip, J.D.; et al. Neutralization of SARS-CoV-2 Omicron variant by sera from BNT162b2 or Coronavac vaccine recipients. Clin Infect Dis 2021, doi:10.1093/cid/ciab1041.
  7. Yu, J.; Collier, A.Y.; Rowe, M.; Mardas, F.; Ventura, J.D.; Wan, H.; Miller, J.; Powers, O.; Chung, B.; Siamatu, M.; et al. Neutralization of the SARS-CoV-2 Omicron BA.1 and BA.2 Variants. N Engl J Med 2022, 386, 1579-1580, doi:10.1056/NEJMc2201849.
  8. Ou, J.; Lan, W.; Wu, X.; Zhao, T.; Duan, B.; Yang, P.; Ren, Y.; Quan, L.; Zhao, W.; Seto, D.; et al. Tracking SARS-CoV-2 Omicron diverse spike gene mutations identifies multiple inter-variant recombination events. Signal Transduct Target Ther 2022, 7, 138, doi:10.1038/s41392-022-00992-2.
  9. Magazine, N.; Zhang, T.; Wu, Y.; McGee, M.C.; Veggiani, G.; Huang, W. Mutations and Evolution of the SARS-CoV-2 Spike Protein. Viruses 2022, 14, doi:10.3390/v14030640.
  10. Fratev, F. R346K Mutation in the Mu Variant of SARS-CoV-2 Alters the Interactions with Monoclonal Antibodies from Class 2: A Free Energy Perturbation Study. J Chem Inf Model 2022, 62, 627-631, doi:10.1021/acs.jcim.1c01243.
  11. Koyama, T.; Miyakawa, K.; Tokumasu, R.; S, S.J.; Kudo, M.; Ryo, A. Evasion of vaccine-induced humoral immunity by emerging sub-variants of SARS-CoV-2. Future Microbiol 2022, 17, 417-424, doi:10.2217/fmb-2022-0025.
  12. McLean, G.; Kamil, J.; Lee, B.; Moore, P.; Schulz, T.F.; Muik, A.; Sahin, U.; Tureci, O.; Pather, S. The Impact of Evolving SARS-CoV-2 Mutations and Variants on COVID-19 Vaccines. mBio 2022, 13, e0297921, doi:10.1128/mbio.02979-21.
  13. Muttineni, R.; R, N.B.; Putty, K.; Marapakala, K.; K, P.S.; Panyam, J.; Vemula, A.; Singh, S.M.; Balachandran, S.; S, T.V.; et al. SARS-CoV-2 variants and spike mutations involved in second wave of COVID-19 pandemic in India. Transbound Emerg Dis 2022, 69, e1721-e1733, doi:10.1111/tbed.14508.
  14. Di Giacomo, S.; Mercatelli, D.; Rakhimov, A.; Giorgi, F.M. Preliminary report on severe acute respiratory syndrome coronavirus 2 (SARS-CoV-2) Spike mutation T478K. J Med Virol 2021, 93, 5638-5643, doi:10.1002/jmv.27062.
  15. Andrews, N.; Stowe, J.; Kirsebom, F.; Toffa, S.; Rickeard, T.; Gallagher, E.; Gower, C.; Kall, M.; Groves, N.; O'Connell, A.M.; et al. Covid-19 Vaccine Effectiveness against the Omicron (B.1.1.529) Variant. N Engl J Med 2022, 386, 1532-1546, doi:10.1056/NEJMoa2119451.
  16. Speletas, M.; Voulgaridi, I.; Sarrou, S.; Dadouli, A.; Mouchtouri, V.A.; Nikoulis, D.J.; Tsakona, M.; Kyritsi, M.A.; Peristeri, A.M.; Avakian, I.; et al. Intensity and Dynamics of Anti-SARS-CoV-2 Immune Responses after BNT162b2 mRNA Vaccination: Implications for Public Health Vaccination Strategies. Vaccines (Basel) 2022, 10, doi:10.3390/vaccines10020316.
  17. Lau, E.H.Y.; Tsang, O.T.Y.; Hui, D.S.C.; Kwan, M.Y.W.; Chan, W.H.; Chiu, S.S.; Ko, R.L.W.; Chan, K.H.; Cheng, S.M.S.; Perera, R.; et al. Neutralizing antibody titres in SARS-CoV-2 infections. Nat Commun 2021, 12, 63, doi:10.1038/s41467-020-20247-4.
  18. Gharbharan, A.; Jordans, C.C.E.; GeurtsvanKessel, C.; den Hollander, J.G.; Karim, F.; Mollema, F.P.N.; Stalenhoef-Schukken, J.E.; Dofferhoff, A.; Ludwig, I.; Koster, A.; et al. Effects of potent neutralizing antibodies from convalescent plasma in patients hospitalized for severe SARS-CoV-2 infection. Nat Commun 2021, 12, 3189, doi:10.1038/s41467-021-23469-2.
  19. Moss, P. The T cell immune response against SARS-CoV-2. Nat Immunol 2022, 23, 186-193, doi:10.1038/s41590-021-01122-w.
  20. Bertoletti, A.; Le Bert, N.; Tan, A.T. SARS-CoV-2-specific T cells in the changing landscape of the COVID-19 pandemic. Immunity 2022, 55, 1764-1778, doi:10.1016/j.immuni.2022.08.008.

Reviewer 2 Report

The paper by Alvarez-Diaz and co-workers reports about the study of in vitro neutralization (NT) antibodies elicited  by vaccination against SARS CoV-2. The Authors performed  the NT against  5 VoC circulating in Colombia. The most relevant weakness point is that the Authors do not report  the number of pastients included in the protocol: this datum is  in some way  obtainab le by  counting the indivdual tables. This reviewer believes that this information must be clearly stated in order to allow the readers to get a precise value of the results. So far, it looks like that the number of samplese /individual studies is very low and consequently it is unrealist to draw any conclusion based on the reported data.In addition Table 2 is somehow misleading since the first lines report demographic information while the remaing are reporting data from the NT testing. The Authors should also consider that "complete vaccination  regimens"  scheme should be declared in the M&M Section: does this  means that all the individuals sampled received two doses? Which was the interval in between the doses? Why  the specimens for NT were collected with a range time of 4 (9-13)  weeks? This is quite a long interval in the evaluation  the development of specific NT antibodies (in other words:  the ab response could differ by far in about one month...).  Why the Authors did not followed  the NT antivobodies development in each individual over the time with more than one sample? The study should be extensively revised including more samples and the missing information.

Author Response

Response to Reviewer 2 Comments

The paper by Alvarez-Diaz and co-workers reports about the study of in vitro neutralization (NT) antibodies elicited  by vaccination against SARS CoV-2. The Authors performed  the NT against  5 VoC circulating in Colombia.

Point 1: The most relevant weakness point is that the Authors do not report the number of patients included in the protocol: this datum is in some way obtainable by  counting the individual tables. This reviewer believes that this information must be clearly stated in order to allow the readers to get a precise value of the results.

Response 1: The number of subjects for each vaccine group were included in the methods section

Manuscript with changes marked (Page 3, Lines 123-129).

Serum samples were collected from immunologically naïve individuals at 13 weeks after receiving the first dose of BNT162b2 (n = 31; 3 males and 28 females, age range, 23–62 years), CoronaVac (n = 30; 9 males and 21 females; age range 18–58 years) and ChAdOx1 (n = 31; 14 males and 17 females; age range 50 –81 years). While samples from patients vaccinated with Ad26.COV2.S (n = 26; 10 males and 16 females; age range 21 –61 years ) were collected 20 weeks after receiving one dose of this vaccine.

Point 2: So far, it looks like that the number of samples/individual studies is very low and consequently it is unrealistic to draw any conclusion based on the reported data.

Response 2: The low number of samples evaluated is indeed a limitation of this study. However, our results are highly comparable with similar studies around the world and published in high impact journals. Some examples indicating the number of samples evaluated from studies already cited in the manuscript are indicated below:

  1. Uriu and Kimura evaluated nAb responses against eight SARS-COV-2 variants in 14 Pfizer vaccinated individuals and 13 individuals who had recovered from Covid-19 [1].
  2. Planas et al. [2] evaluated neutralizing activity of sera from recipients of the Pfizer vaccine, sampled at week 3 (W3) after vaccination (n = 16) and week 8 (W8) after vaccination (week 5 after the second dose), (n = 16). Neutralizing activity of sera from recipients of the AstraZeneca vaccine, sampled at week 10 (W10) after vaccination (n = 23) and week 16 (W16) after vaccination (week 4 after the second dose) (n = 20).
  3. Wang et al. [3], compared EC50 values of neutralization against alpha, beta, gamma and Delta variants in 15 individuals vaccinated with Pfizer.
  4. Lopera et al [4], evaluated nAb responses using PRNT assays in groups of sixty individuals after 30 days (n=60), 90 days (n=60), and 180 days (n=56) (4 donors were excluded because of SARS-CoV-2 infection during the follow-up).
  5. Yu et al [5], measured neutralizing antibody titers that by a luciferase-based pseudovirus neutralization assay in serum samples obtained from 24 persons who had been vaccinated and boosted with Pfizer.

Nonetheless, the low number of samples is a limitation, and this was highlighted in the discussion: Manuscript with changes marked (Page 10, Lines 321-326).

Altogether, the evidence points to considerable drops in both nAb responses and vaccine efficacy against SARS-CoV-2 emerging variants with a high impact on public health. However, this study has several limitations, including a small sample size with demographic heterogeneity, the absence of samples from the pediatric or adolescent population, and a lack of information on the contribution of B or T cell responses against the variants studied.

Point 3: In addition, Table 2 is somehow misleading since the first lines report demographic information while the remaining are reporting data from the NT testing. The Authors should also consider that "complete vaccination regimens” scheme should be declared in the M&M Section: does this means that all the individuals sampled received two doses? Which was the interval in between the doses?

Response 3: Demographic information was excluded from Table 2 and explanation of vaccination schemes was included in the method section. Manuscript with changes marked (Page 3, Lines 113-119).

This was a non-probabilistic, consecutive cross-sectional study of four Colombian cohorts included in the National Vaccination Plan during the prioritization phase [6]. The first three cohorts received a vaccination schedule of two doses with four weeks interval of BNT162b2 (Pfizer), ChAdOx1 (AstraZeneca), or CoronaVac (Sinovac). The fourth cohort consisted of vaccinated with one dose of Ad26.COV2.S (Janssen). Due to the limited availability of individuals vaccinated with mRNA-1273 (Moderna) at the sampling time, this cohort was not included in the study.

Point 4: Why the specimens for NT were collected with a range time of 4 (9-13) weeks? This is quite a long interval in the evaluation the development of specific NT antibodies (in other words:  the ab response could differ by far in about one month...). 

Response 4: There were typing errors in the following times indicated in the first version of this manuscript. The right times were 13 weeks after receiving the first dose of BNT162b2, CoronaVac and ChAdOx1, while 20 weeks after receiving one dose of Ad26.COV2.S

Manuscript with changes marked (Page 3 Lines 123-129):

Serum samples were collected from immunologically naïve individuals at 13 weeks after receiving the first dose of BNT162b2 (n = 31; 3 males and 28 females, age range, 23–62 years), CoronaVac (n = 30; 9 males and 21 females; age range 18–58 years) and ChAdOx1 (n = 31; 14 males and 17 females; age range 50 –81 years). While samples from patients vaccinated with Ad26.COV2.S (n = 26; 10 males and 16 females; age range 21 –61 years ) were collected 20 weeks after receiving one dose of this vaccine.

Point 5: Why the Authors did not followed the NT antibodies development in each individual over the time with more than one sample? The study should be extensively revised including more samples and the missing information.

Response 5: Although the nAb responses over time would contribute with valuable information this was not the scope of this work. Instead, we addressed to evaluate the nAb responses elicited by the initial regime of the main SARS-CoV-2 vaccines administered in the country, against Gamma, Mu, Delta, and Omicron which were the variants with higher impact on public health in Colombia. In addition, this is the first study that compare nAb responses between these vaccines and variants.

REFERENCES

  1. Uriu, K.; Kimura, I.; Shirakawa, K.; Takaori-Kondo, A.; Nakada, T.A.; Kaneda, A.; Nakagawa, S.; Sato, K.; Genotype to Phenotype Japan, C. Neutralization of the SARS-CoV-2 Mu Variant by Convalescent and Vaccine Serum. N Engl J Med 2021, 385, 2397-2399, doi:10.1056/NEJMc2114706.
  2. Planas, D.; Veyer, D.; Baidaliuk, A.; Staropoli, I.; Guivel-Benhassine, F.; Rajah, M.M.; Planchais, C.; Porrot, F.; Robillard, N.; Puech, J.; et al. Reduced sensitivity of SARS-CoV-2 variant Delta to antibody neutralization. Nature 2021, 596, 276-280, doi:10.1038/s41586-021-03777-9.
  3. Wang, B.; Goh, Y.S.; Fong, S.W.; Young, B.E.; Ngoh, E.Z.X.; Chavatte, J.M.; Salleh, S.N.M.; Yeo, N.K.; Amrun, S.N.; Hor, P.X.; et al. Resistance of SARS-CoV-2 Delta variant to neutralization by BNT162b2-elicited antibodies in Asians. Lancet Reg Health West Pac 2021, 15, 100276, doi:10.1016/j.lanwpc.2021.100276.
  4. Lopera, T.J.; Chvatal-Medina, M.; Florez-Alvarez, L.; Zapata-Cardona, M.I.; Taborda, N.A.; Rugeles, M.T.; Hernandez, J.C. Humoral Response to BNT162b2 Vaccine Against SARS-CoV-2 Variants Decays After Six Months. Front Immunol 2022, 13, 879036, doi:10.3389/fimmu.2022.879036.
  5. Yu, J.; Collier, A.Y.; Rowe, M.; Mardas, F.; Ventura, J.D.; Wan, H.; Miller, J.; Powers, O.; Chung, B.; Siamatu, M.; et al. Neutralization of the SARS-CoV-2 Omicron BA.1 and BA.2 Variants. N Engl J Med 2022, 386, 1579-1580, doi:10.1056/NEJMc2201849.
  6. Malagon-Rojas, J.; Mercado-Reyes, M.; Toloza-Perez, Y.G.; Galindo, M.; Palma, R.M.; Catama, J.; Bedoya, J.F.; Parra-Barrera, E.L.; Meneses, X.; Barbosa, J.; et al. Comparison of Anti-SARS-CoV-2 IgG Antibody Responses Generated by the Administration of Ad26.COV2.S, AZD1222, BNT162b2, or CoronaVac: Longitudinal Prospective Cohort Study in the Colombian Population, 2021/2022. Vaccines (Basel) 2022, 10, doi:10.3390/vaccines10101609.

Reviewer 3 Report

Alvarez-Diaz et al compared SARS-CoV-2 IgG responses between 9-13 weeks following vaccination with 2 doses of either BNT162b2, CoronaVac, ChAdOx1-s or Ad26.CoV2.S in individuals naïve for SARS-CoV-2 infection in Colombia. The authors measured IgG binding using ELISA and neutralising responses using microneutralisation assay. They found that BNT162b2 induced the highest SARS-CoV-2 S-specific IgG titers compared to other vaccines. They observed that neutralising antibody titers against Omicron induced by BNT162b2 and Ad26.CoV2.S did not correlate with S-specific IgG binding. Similar results were observed with neutralising antibodies against Delta induced by BNT162b2. Globally, the authors measured a reduction in neutralising responses against Mu, Delta and especially Omicron induced by all vaccines compared to B.1.111 (ancestral strain) and Gamma.

It is relevant to compare antibody responses induced by a range of vaccines side-by-side. It has been done in different countries and it is interesting to compare the results. The methods look appropriate to me. The figures are clear and the manuscript is well written. Please find below minor comments:

Minor comments:

1) Lines 84-87: it is very difficult to read this sentence and understand the percentages. Maybe it would be clearer to split the sentence into two sentences.

2) Table 2: The sub cohorts which received different vaccines are very heterogenous (age, sex). The authors should mention it as limitation or potential bias.

3) Line 184: typo ? Should it be “the lowest” ?

3) Line 253: a role of T cell responses in protection against severe disease should be mentioned. 

4) Line 275: In addition to surveillance, the authors could mention the importance to develop novel improved vaccine strategies leading to a better cross-protection and mucosal immunity.

5) Figure 1: it looks antibody titers after BNT162b2 vaccination could be even higher. The ELISA looks slightly saturated. Could the authors repeat these specific samples with higher dilutions ?

6) Discussion: A difference in neutralising responses against variants were observed between the vaccines. Could the authors discuss hypotheses ? For example it looks adenovirus-based vaccines are better to induce cross-reactive neutralising responses compared to BNT162b2.

7) The authors mention they confirmed the absence of anti-S before vaccination and anti-N during clinical follow-up to confirm individuals were not infected. I think it would be relevant to show this data as supplementary data.

Author Response

Please find the responses in the attached file

Reviewer 4 Report

The paper by Alvarez-Diaz is an original research paper describing the antibody response to four of the anti-Covid19 vaccines in the population in Colombia.
The introduction is sufficiently informative. The methods section is quite good, but a table with subjects information and grouping is highly welcome.
The results and discussion section are ok, and the study limts are stated in the text. Probably, adding data from booster sera would have much increased the significance of the content.
A major issue with the paper is that it republishes some results already used for a previous publication of the same authors, here cited as ref 4 (and 16), which studies the cohort of subjects
immunized with Pfizer vaccine. Double publication of the same results is not a policy I encourage, so I strongly suggest to make use of othe samples and repeat testing.
Some minor points:
- line 109: please provide the number of subjects (total and divided for each vaccine group)
- line 111-112: since the administration of 5 vaccines has been mentioned in the previous section, please explain how only 4
have been choosed for the study
- line 134: have viral isolates actually been used for the assays, or rather viral stocks derived from isolates through
passaging in cell cultures (which is more commonly performed)?
- line 149: the paragraph is incomplete; e.g.: both fig1 and tab1 make use of statistycal tests which are not mentioned here
- line 167: in fig.1, it is not very clear what the histograms depict (maybe the geometric mean? it should be indicated somewhere).
In any case, it looks as the results are underestimate for the pfizer group (blue), as there are many samples with Abs levels at the
top edge of the graph. These should have been rerun after further dilution to be able to obtain amore accurate calculation of the group's
mean titre
- line 168: "*: p <0.01"
- line 193: In fig2, for the Pfizer vaccine group, the difference between the B.1 and the P.1 titre is marked as n.s, which looks
unlikely watching at the strong and neat difference shown in the graph. Please check.
- line 197: "*: p <0.01"
- line 225-227: the sentence lacks a principal verb
- line 232-234: please rephrase in a more correct form
- line 248-250: please rephrase in a more correct form
- line 276: please check for authors' contributions. Looks like there are some which only contributed resources, which
is not usually recognised as a sufficient authorship criterium

- line 303: please check if reference style is in accordance with journal's guidelines
- line 306: please check if reference style is in accordance with journal's guidelines
- line 318: please check if reference style is in accordance with journal's guidelines
- line 345: please check if reference style is in accordance with journal's guidelines
- line 349: ref 16 is the same as ref 4
- line 352: please check if reference style is in accordance with journal's guidelines
- line 358: please check for issue and doi information
- line 362: please check for reference title correctness
- line 380-383: the referenced papers are actually the same one. Please cite only ref 27.

Author Response

(The authors gave the same response as above.)

Round 2

Reviewer 4 Report

I thank the authors for modifying their paper following many of my suggestions.

Two of my concerns from the previous round of revision still remain:

- The use of data from the previous publication, which is in my humble opinion an incorrect choice that many other groups, working in the same conditions, have choosen to avoid. In case the authors have no intentions to provide new data, making use of the samples they have, it'll be necessary that they clearly state in both the methods, results, and discussion section which of the presented data come from the previous study, with the appropriate citation, and which ones are original for this one, so that the readers may easily distinguish what is new results from what is background data.

- The minor point about authorship received an unclear answer. The criteria for authorship for this journal are clearly explained here:  https://www.mdpi.com/journal/vaccines/instructions#authorship

https://www.icmje.org/recommendations/browse/roles-and-responsibilities/defining-the-role-of-authors-and-contributors.html#two

By the way, one of the authors has her name changed from Tatiana to Yussely in this section. Please check.

I will consider the paper for publication after a second revision focused especially to the first point, but I would also like the authors to be taking into consideration the second as well. Authorship is important and should be managed properly. Having almost twenty co-authors for a research article has become a common fashion in recent years but it might be unfair from many points of view.

Author Response

Response to Reviewer 4 Comments

Reviewer comments:

I thank the authors for modifying their paper following many of my suggestions.

Two of my concerns from the previous round of revision still remain:

Point 1: The use of data from the previous publication, which is in my humble opinion an incorrect choice that many other groups, working in the same conditions, have choosen to avoid. In case the authors have no intentions to provide new data, making use of the samples they have, it'll be necessary that they clearly state in both the methods, results, and discussion section which of the presented data come from the previous study, with the appropriate citation, and which ones are original for this one, so that the readers may easily distinguish what is new results from what is background data.

Response 1: Unfortunately we have limited samples with these characteristics, hence we have to use the same samples to include new data from nAb responses against Delta and Omicron. Hence, following the advice of the reviewer, we clearly state this in the methods, results (tables, figures, and body text), and discussion sections.

In methods: Manuscript with changes marked in green (Page 3, Line 126-129).

…For the sample sub-set of BNT162b2 vaccinated individuals, we used the same patients’ sera and data used previously to compare preliminary results of nAb responses against B.1.111, Mu, and Gamma with new data on different vaccines and SARS-CoV-2 variants [4].

In results (Table 1): Manuscript with changes marked in green (Page 5, Line 189-190).

a: binding and neutralizing antibody titers from BNT162b2 vaccinated individuals were reported previously [4], and re-analyzed considering a limit of detection of 1:10.

In results (Figure 2): Manuscript with changes marked in green (Page 7, Line 219-220).

a: data of nAb titers from BNT162b2 vaccinated individuals were reported previously [4], and re-analyzed considering a limit of detection of 1:10.

In results (Table 2): Manuscript with changes marked in green (Page 8, Line 222-223).

. a: data from nAb titers from BNT162b2 vaccinated individuals were reported previously [4], and re-analyzed considering a limit of detection of 1:10.

In Discussion: Manuscript with changes marked in green (Page 8, Line 233-236).

We previously reported the neutralizing responses of BNT162b2 vaccinated individuals against Mu and Gamma [4]. However, a more complete picture of vaccine-elicited nAb responses against the main SARS-CoV-2 variants in Colombia.

Point 2: The minor point about authorship received an unclear answer. The criteria for authorship for this journal are clearly explained here:  https://www.mdpi.com/journal/vaccines/instructions#authorship

https://www.icmje.org/recommendations/browse/roles-and-responsibilities/defining-the-role-of-authors-and-contributors.html#two

By the way, one of the authors has her name changed from Tatiana to Yussely in this section. Please check.

I will consider the paper for publication after a second revision focused especially to the first point, but I would also like the authors to be taking into consideration the second as well. Authorship is important and should be managed properly. Having almost twenty co-authors for a research article has become a common fashion in recent years but it might be unfair from many points of view.

Response 1: Thank you for this wise recommendation regarding authorship management. We revised thoroughly the author’s contributions and decided to modify the author list following the MDPI Change of Authorship Form.

Round 3

Reviewer 4 Report

I thank the authors for further modifying their paper following my indications, I conseder the paper to be acceptable for publication in this form